# Cutting-Edge Sensor Design: MIP Nanoparticle-Functionalized Nanofibers for Gas-Phase Detection of Limonene in Predictive Agriculture

**DOI:** 10.3390/polym17030326

**Published:** 2025-01-25

**Authors:** Fabricio Nicolàs Molinari, Marcello Marelli, Enrico Berretti, Simone Serrecchia, Roxana Elisabeth Coppola, Fabrizio De Cesare, Antonella Macagnano

**Affiliations:** 1Institute of Atmospheric Pollution Research (IIA)-CNR, 00010 Montelibretti, RM, Italy; fmolinari@inti.gob.ar (F.N.M.); simoneserrecchia@cnr.it (S.S.); decesare@unitus.it (F.D.C.); 2National Institute of Industrial Technology (INTI), Buenos Aires B1650WAB, Argentina; rcoppola@inti.gob.ar; 3Institute of Science and Chemical Technologies “Giulio Natta” (SCITEC)-CNR, 20138 Milano, MI, Italy; marcello.marelli@scitec.cnr.it; 4Institute for the Chemistry of OrganoMetallic Compounds (ICCOM)-CNR, 50019 Sesto Fiorentino, FI, Italy; enrico.berretti@cnr.it; 5Department for Innovation in Biological, Agrofood and Forest Systems (DIBAF), University of Tuscia, 01100 Viterbo, VT, Italy

**Keywords:** molecular imprinted polymers, BVOCs, monoterpenes, electrospinning, nanoparticles, nanofibers, conductive sensors, limonene, predictive agriculture

## Abstract

As population growth and climate change intensify pressures on agriculture, innovative strategies are vital for ensuring food security, optimizing resources, and protecting the environment. This study introduces a novel approach to predictive agriculture by utilizing the unique properties of terpenes, specifically S(-)-limonene, emitted by plants under stress. Advanced sensors capable of detecting subtle limonene variations offer the potential for early stress diagnosis and precise crop interventions. This research marks a significant leap in sensor technology, introducing an innovative active sensing material that combines molecularly imprinted polymer (MIP) technology with electrospinning. S(-)-limonene-selective MIP nanoparticles, engineered using methacrylic acid (MAA) and ethylene glycol dimethacrylate (EGDMA), were synthesized with an average diameter of ~160 nm and integrated into polyvinylpyrrolidone (PVP) nanofibers reinforced with multiwall carbon nanotubes (MWCNTs). This design produced a conductive and highly responsive sensing layer. The sensor exhibited rapid stabilization (200 s), a detection limit (LOD) of 190 ppb, and a selectivity index of 73% against similar monoterpenes. Optimal performance was achieved at 55% relative humidity, highlighting environmental conditions’ importance. This pioneering use of polymeric MIP membranes in chemiresistive sensors for limonene detection opens new possibilities for monitoring VOCs, with applications in agricultural stress biomarkers, contaminant detection, and air quality monitoring, advancing precision agriculture and environmental protection.

## 1. Introduction

With a growing global population and climate change intensifying pressures on agriculture, innovative strategies are critical to boost crop productivity by 70% by 2050 [1]. Ensuring food security while conserving resources and protecting the environment requires improving nutrient uptake and use efficiency (NUE) [2] and reducing crop losses caused by stress, for example, through pests and phytopathogen control agents [3,4,5]. Sustainable solutions include eco-smart biocontrol agents [6], crop genetic resistance to pathogens [7], and enhancing the tolerance to drought, high temperatures, waterlogging, and soil salinization [8], respectively. These approaches are essential for meeting future food demands while mitigating climate impacts. Monitoring crop health and detecting early signs of plant nutrient deficiency and plant stress, biotic and abiotic, is an approach to improving or preserving yields by preventing crop losses [9].

Several studies have explored methods to monitor adverse events affecting crops, but conventional approaches are often invasive, costly, and time-consuming. These methods rely on sample collection, laboratory processing, and destructive procedures, which can trigger stress responses and fail to provide accurate spatial data [10].

Recently, non-destructive technologies have emerged as a promising alternative. While sometimes less precise, they offer significant advantages, such as being non-invasive, providing ecosystem-wide data, and covering larger areas, from individual plants to entire fields [11]. These tools can detect crop health changes, such as nutrient deficiencies or diseases, and quantify symptom severity, enabling more efficient application of treatments like fertilizers or pest control agents. This aligns with precision agriculture goals to optimize resource use and reduce environmental impacts [12]. However, these sensing technologies often require careful calibration and tailored sampling systems to achieve reliable results.

Remote sensing is among the most widely used non-invasive and non-destructive technologies for monitoring plant parameters over large areas and across temporal and spatial scales [11]. Advanced tools, such as portable Raman-based sensors, can detect stress-related metabolites, reducing agrochemical use and optimizing yields [13]. Additionally, electronic noses (e-noses) are gaining traction for detecting plant-released volatile organic compounds (VOCs), helping to identify nutrient deficiencies [14], assess fruit maturity [15], and detect pathogen attacks [16]. Most of the efforts in monitoring plant health, anyway, have been carried out to obtain an early and accurate detection of biotic stressors, such as pathogens and pests [17], and abiotic stressors like drought, salinity, temperature changes, and light [18]. These approaches enable timely interventions to mitigate yield losses and improve crop management.

Volatile organic compounds (VOCs) are chemicals released into the atmosphere from both human activities and natural processes, such as those in soil, oceans, and vegetation. Biogenic VOCs (BVOCs), defined as organic trace gases excluding carbon dioxide, carbon monoxide, and methane [19], include isoprenoids (isoprene and monoterpenes), alkanes, alkenes, carbonyls, alcohols, esters, ethers, and acids. Terrestrial vegetation produces around 90% of global VOC emissions [20]. Plants emit thousands of VOC species (PVOCs), mostly isoprene (C_5_H_8_) and monoterpenes (C_10_H_16_) as the most representative (70% and 11%, respectively), followed by sesquiterpenes (C_15_H_24_), methanol (CH_3_OH), and acetone (CH_3_COCH_3_) [21].

Terpenes are emitted by plants in response to abiotic stresses [22] but also upon phenological statuses, like fruit ripening, plant age, and organs [23,24]. Acting as signaling molecules, terpenes facilitate plant interactions with bacteria, fungi, and insects, playing roles in defense and in both antagonistic and mutualistic relationships (biotic stresses) [25]. For instance, limonene, a compound released by plants, can serve as a valuable biomarker for monitoring plant health and detecting early signs of stress or disease. In grapevines, limonene is crucial during nematode attacks and in response to water stress [26,27]. Similarly, in olive trees, limonene emissions are associated with fruit ripening and act as a repellent against the olive fruit fly (*Bactrocera oleae* (Rossi)) during oviposition [28,29]. This opens up opportunities to develop selective sensors tailored to specific terpenes. Therefore, developing selective sensors to monitor limonene emissions as well as selected monoterpenes could provide precise diagnostics, directly tied to the stressors impacting plant health.

Nonetheless, the class of monoterpenes remains relatively underrepresented in terms of sensor development, especially when compared to other classes of VOCs. To the best of our knowledge, there has been limited advancement in the development of chemical sensors, particularly for the selective detection of monoterpenes, and even fewer commercial sensors dedicated to this purpose, except for GC-MS technologies and commercial electronic noses based on metal oxide, electrochemical sensors, and conducting polymers [30]. While the electronic nose offers portability, it lacks sensitivity and precise quantification, requiring data analysis and frequent calibration, whereas GC-MS provides accurate identification and quantification, but it is costly, complex, and less practical for real-time field monitoring in agriculture. Recently, customized sensors, based on their electrical signals, such as chemiresistors, electrochemical sensors, and chemicapacitors, have made significant advancements for monitoring VOCs and terpenes in plant diseases [30]. Appendix A groups some of the most recent advancements in sensors for VOCs in agriculture. Each sensor technology reports its own strengths, such as high sensitivity or specificity, but also faces limitations related to environmental interference, stability, and selectivity. For instance, chemiresistor sensors offer high sensitivity and are cost-effective but lack specificity in complex environments [30]. Electrochemical sensors excel in sensitivity and selectivity, but they are subjected to consumption, then short lifespans. Gravimetric sensors, though unique in focusing on mass changes, generally offer lower sensitivity and are affected by environmental conditions, requiring improvements for reliable field use. On the other hand, colorimetric sensors can detect specific VOCs by changing color, creating visual fingerprints that can be analyzed with a camera [30]. While simple and useful in agriculture, they suffer from low sensitivity and are affected by environmental factors like humidity and light. To improve these, researchers are investigating the effects of the addition of metal- and covalent-organic frameworks (MOFs and COFs), which should enhance sensitivity and environmental resistance [31]. Bioluminescent sensors using genetically modified bacteria to emit light in response to VOCs have been employed to detect early disease in fungal infections in oranges before visible symptoms appear. However, maintaining bacterial activity over time remains a challenge, and real-world performance needs improvement [32].

SERS sensors amplify Raman scattering to detect VOCs with high sensitivity. While effective for detecting plant VOCs, weak interactions between sensor materials and VOCs can limit robustness. Nonetheless, the combination with nanomaterials decreased the low detection limit and demonstrated reversibility, suitable for continuous monitoring [33]. Wearable sensors, fabricated using flexible materials, offer non-invasive, real-time monitoring of plant VOCs. These small, flexible sensors detect health indicators like methanol emissions in plants, providing continuous data in the field. Though promising for real-time monitoring, their durability in diverse environmental conditions needs improvement [34,35]. LSPR sensors detect target molecules via oscillations in metallic nanoparticles. The addition of molecular imprinting polymer (MIP) technology improves selectivity but can reduce sensitivity [36]. For instance, a sensor with gold nanoparticles showed a 12.33 times higher sensitivity for detecting cis-jasmone, making it useful for plant health monitoring and pathogen detection despite the trade-off in sensitivity [37].

Therefore, the gap in the current sensor landscape presents an opportunity for exploiting the potentials of the molecularly imprinted polymers (MIPs). Indeed, as of today, the most performing sensors for terpenes are mainly based on MIP technology.

MIPs replicate the highly specific molecular recognition process found in natural biological macromolecules, such as enzyme–substrate or antibody–antigen interactions, by utilizing tailored interactions to recognize specific ligands [38]. The MIP synthesis is based on the creation of template-induced formation of complementary recognition cavities within a polymer matrix, shaped for a particular template molecule, that then result in a specific shape, size, and functionality for the target chemical or biological molecule used as the template. During polymerization, monomers and crosslinkers assemble around the template molecule, which is subsequently removed, leaving behind a highly specific and complementary empty active site. As a result, MIPs exhibit highly selective interactions with target analytes and provide greater chemical and thermal stability compared to biological systems, owing to the high degree of crosslinking in the polymer matrix [39,40].

MIPs bind analytes through non-covalent interactions, such as hydrogen bonding, π–π interactions, and van der Waals forces, offering high discrimination and precise quantification. These non-covalent interactions enable repeated binding and release of VOCs, ensuring durability across multiple uses. Moreover, MIP-based sensors exhibit superior chemical and thermal stability compared to biomolecules, providing reliable performance under various conditions [39]. MIP technology has been widely used to develop advanced selective sensors for VOC detection. Based on such technology, these sensors incorporate tailored binding sites that precisely match the molecular configurations of VOCs, like terpenes. The MIP cavities function through a lock-and-key mechanism, utilizing size, shape, or functional group compatibility [41].

The design of MIP-based materials for sensors typically follows two main approaches: top-down and bottom-up.

The top-down strategy involves creating a bulk material with specific cavities, which is then ground down to achieve micro- or nano-scale dimensions. In contrast, the bottom-up approach involves direct polymerization of nanoparticles, eliminating the grinding step and simplifying the production process.

Over time, MIP development has employed various methods, including bulk, precipitation, and emulsion polymerization, as well as sol-gel techniques and electro-polymerization [42,43].

Nanostructures are often preferred due to their high surface-area-to-volume ratio, which provides more accessible binding sites, enhances interaction with target analytes, and improves sensitivity and specificity.

Nanostructured MIP-based VOC sensors have been developed in various forms, including chemiresistors, piezoelectric sensors, and optical sensors [44]. Among them, MIP-based chemiresistors have demonstrated greater versatility in fabrication and superior sensing performance [44]. In these sensors, changes in electrical resistance or conductivity occur in response to specific chemical compounds, allowing for their detection and quantification. Such performances have been further significantly enhanced by combining MIPs with conductive polymers (e.g., polyaniline, polypyrrole, and poly(3-hexylthiophene)) [45], carbon-based nanomaterials (e.g., nanotubes, nanopowders, and graphene), or metals/metal oxides [46,47,48,49,50]. These combinations enable the creation of highly specific VOC binding sites, improving sensor selectivity and reducing interference. These interactions significantly alter the electrical properties of the conductive polymer, such as charge distribution or morphology. The integration of molecularly imprinted polymers (MIPs) with conductive materials results in highly effective sensors, known as c-MIPs. These sensors are highly specific and selective for VOCs due to the tailored MIP binding sites. They also exhibit increased sensitivity, as the MIPs’ pores and cavities enhance the surface area and target molecule interaction. Furthermore, the imprinting process pre-organizes charge distribution, improving conductivity and capacitance. Lastly, these sensors are less susceptible to interference from other substances, further enhancing their reliability.

Additionally, the robustness of MIPs ensures stability and consistent performance under diverse environmental conditions. The versatility of this approach allows for the customization of both the conductive polymer and the MIPs, tailoring them to desired electrical properties and specific VOC recognition, respectively.

Following the importance of monitoring terpene release in the atmosphere, such as for limonene, for agricultural purposes, and the possibility of developing highly selective MIP-based sensors, a series of studies have highlighted the effectiveness of MIP-based limonene sensors. In Section 3.5, Table 3 summarizes some features of the sensors focused on limonene (and monoterpenes) employing MIP technology. Depending on the molar ratio among the components, the porogen solvents used, and the final architecture of the sensing devices, the range of limonene detection changed [51,52,53,54,55,56,57,58,59], as did the sensitivity and selectivity. However, it was observed that while MIPs were designed to be highly selective and capable of linear responses at high concentrations, their sensitivity could fall short when detecting the trace levels of VOCs emitted by plants, particularly under variable environmental conditions. Therefore, a more robust and sensitive MIP sensor for terpenes, and specifically limonene, should be designed for these types of applications.

Among the various techniques for designing performing sensors, electrospinning is a versatile, scalable, and cost-effective electrodynamic method that enables the production of both microfibers and nanofibers, which can be adapted to different substrates for sensor design and integration. Additionally, electrospinning supports a broad range of polymers, including eco-friendly options, thus promoting sustainability in sensor production [60,61]. The advantages of incorporating nanoscale features from electrospun fibers in high-performance sensor fabrication have been extensively documented [62,63,64,65]. Thus, combining molecularly imprinted polymer (MIP) technology with electrospinning (ES) holds significant potential for developing advanced chemosensors for VOC recognition [66]. Nanofibrous structures incorporating MIPs enable the creation of more complex architectures, improving their properties compared to bulk matrices or thin films [66,67,68,69,70]. Thus, a MIP-nanofibrous (MIP-NF) sensor, which combines molecularly imprinted polymers (MIPs) with electrospinning (ES), offers significant advantages. It operates at room temperature, provides enhanced selectivity, and can be easily customized for specific agricultural applications.

On the other hand, integrating MIP and ES technologies poses challenges due to their different processing methodologies; thus, ensuring compatibility between these techniques is crucial. One prominent approach involves embedding MIP nanoparticles within electrospun fibers, demonstrating promising results for VOC detection [71,72,73,74,75].

In a preliminary study, the authors presented the design of S(-)-limonene-templated molecularly imprinted nanoparticles (MINPs) embedded within polyvinylpyrrolidone (PVP) nanofibers alongside multiwall carbon nanotubes (MWCNTs) to create a selective and conductive sensor [76]. In the present work, a fine-tuning of the parameters of each process enabled the optimization of final architectures and enhancement of recognition performance. While this method deviates from the conventional requirements of QCM sensors, which typically prioritize thin, uniform coatings to minimize viscoelastic damping, it offers significant advantages. It surpasses the sensitivity of traditional bulk MIP coatings and introduces a customizable approach to sensor design.

The proposed strategy was also compared to a previously described approach by the authors [77], where pre-formed polymers (polyacrylic acid (PAA) and polyvinylpyrrolidone (PVP)) were used instead of monomers, and functional cavities for limonene were imprinted directly within the fibers through a UV-crosslinking process. In the present study, the incorporation of MIP nanoparticles into nanofibers was expected to enhance the sensing features, such as a higher number of interaction sites, improved robustness to the fibers, and increased sensitivity due to the tailored recognition sites within the MIP nanostructures. To the best of our knowledge, the use of polymeric MIP nanofibrous membranes as active materials in chemiresistive sensors for limonene detection remains largely unexplored, and this innovative approach offers significant potential for detecting a diverse range of molecules, including biomarkers, contaminants, and air pollutants in the gas phase.

## 2. Materials and Methods

### 2.1. Materials

Polyvinylpyrrolidone (PVP; Mw 1,300,000, Mw 30,000), methacrylic acid (MAA; Mw 86.09), 2-2′-azobis(2-methyl-propionitrile) (AIBN; Mw 164.21), ethylene glycol dimethacrylate (EGDMA; Mw 198.22), multiwalled carbon nanotubes (MWCNT; >99% carbon basis, O.D. × length 6–13 nm × 2.5–20 μm), S(-)-limonene (S-Lim; 96%, Mw 136,23), α-pinene (α-Pin; 98%, Mw 136.23), linalool (Lin; 97%, Mw 154,25), absolute ethanol (EtOH; ≥99.8%, ACS reagent, Mw 46.07), acetonitrile (MeCN; ≥99.5%, ACS reagent, Mw 41.05), and N-N-dimethylformamide (DMF; 99.8% ACS reagent, Mw 73.09) were purchased from Merck KGaA (Darmstadt, Germany) and used without further purification.

### 2.2. Molecularly Imprinted Nanoparticles (MIP-NPs)

Molecularly imprinted nanoparticles (MIP-NPs) were synthesized using methacrylic acid (MAA) as the monomer, ethylene glycol dimethacrylate (EGDMA) as the crosslinker, azobisisobutyronitrile (AIBN) as the free radical initiator, and acetonitrile as the porogen solvent. The effect of the monomer-to-crosslinker ratio on the final polymer properties is well documented [78]. Additionally, the monomer-to-solvent ratio is crucial for obtaining nanoparticles. This study used S(-)-limonene (S-Lim) as the template molecule, with an S-Lim:MAA:EGDMA molar ratio of 1:4:8, respectively. AIBN was used at 2 wt.% of the total reaction mixture. Polymerization was carried out in a round-bottom flask containing 50 mL of acetonitrile following a volume ratio percentage (*v*/*v*) of 0.16:0.30:1.47:98.15 (S-Lim:MAA:EGDMA:acetonitrile) under a nitrogen atmosphere. A solution of 0.5 mmol of S(-)-limonene, 2 mmol of MAA, and 4 mmol of EGDMA was stirred magnetically for 5 min. The dispersion was then sonicated for 20 min using an ultrasonic probe sonicator (Sonics and Materials Inc., Newtown, CT, USA) set at 15% amplitude, followed by an additional 20 min cooling by magnetic stirring. Once the temperature dropped below 25 °C, AIBN was added, and the mixture was purged with nitrogen for 15 min to minimize oxygen interference during polymerization. The polymerization was conducted at 60 °C for 6 h, then the mixture was cooled and stored at 5 °C. The formation of the nanoparticles was indicated by a color change in the solution, from colorless to milky white. Upon stopping the agitation, precipitation occurred at the bottom of the nanoparticle suspension. Non-imprinted nanoparticles (NIP-NPs) were synthesized following the same procedure but without the template molecule. Finally, the entrapped S(-)-limonene molecules were removed from the imprinted nanoparticles through four cycles of washing with cyclohexane and centrifugation at 5500 rpm. The limonene extracted after each centrifugation wash with cyclohexane was measured using the Trace 1310 GC/TSQ 8000 Evo (Thermo Fisher Scientific, Norristown, PA, USA). This procedure confirmed that after the first two washes, the limonene was almost completely extracted from the MIPs.

### 2.3. MWCNT Dispersion

A dispersion of MWCNTs (0.7 wt.%) in DMF was prepared by alternating between ultrasonic probe sonication set at 30% amplitude and magnetic stirring, with PVP K30 used to enhance the stability of the dispersion, following a previously reported method [79].

### 2.4. Electrospinning Solution

Electrospinning solutions containing MIP-NPs were prepared by suspending 10 mg of MIP-NPs in 1 mL of the MWCNT dispersion, followed by 25 min of ultrasonic probe sonication set at 10% amplitude. The resulting MIP/MWCNT dispersion was mixed with 4 mL of a 12 wt.% PVP/EtOH solution (Mw 1.3 KDa) under magnetic stirring and further sonicated for 25 min in an ultrasound bath. The non-imprinted polymer (NIP) solution was prepared using the same procedure, but with NIP-NPs instead of MIP-NPs.

### 2.5. Electrospinning Conditions and Device Fabrication

Electrospinning deposition was carried out using a Fluidnatek^®^ LE-50 machine (Bioinicia, Paterna, Valencia, Spain). The distance between the needle tip and the collector was optimized through testing to produce uniform, dry fibers, with 14 cm determined as the optimal distance. The solution flow rate was 200 µL/h for MIP and NIP nanofibers (RH: 35%; T: 20 °C). The setup included two high-voltage sources: +8 kV applied to the needle and −2 kV to the collector. The deposition time was fixed at 3 min to achieve a homogeneous coating of the transducer microelectrodes connected to the collector using metallic tape.

### 2.6. Interdigitated Electrodes (IDEs)

Interdigitated electrodes (IDEs), supplied by Micrux Technologies (Gijón, Spain), were fabricated on a borosilicate substrate (dimensions: 10 × 6 × 0.75 mm). The Pt/Ti electrodes featured 120 pairs with dimensions of 10 μm in width, 3500 μm in length, and 150 nm in thickness, with a 10 μm gap between them. Before use, the surface of the IDEs was cleaned by rinsing with soap, followed by immersion in a “base piranha” solution (3:1 *v*/*v* mixture of ammonia and hydrogen peroxide) at 60 °C for approximately 60 min, under a well-ventilated fume hood. The electrodes were then thoroughly rinsed with Milli-Q water (~18 MΩ·cm), washed with isopropyl alcohol, and dried with a nitrogen stream.

### 2.7. UV-Crosslinking Process

PVP nanofibers housing MIP- and NIP-NPs were irradiated for 10 min by a 180 nm-to-visible spectrum range-emitting 500 W UV lamp (Polymer Helios Italquartz, Cambiago, MI, Italy). UV irradiation induced crosslinking within the PVP matrix, creating a more interconnected, hydrophobic, and robust polymer network [77,80].

### 2.8. Scanning Electron Microscopy (SEM)

Fiber morphological analyses were carried out using scanning electron microscopy (SEM) micrographs. The electrospun nanofibrous fabrics deposited on thin SiO_2_ wafers and sputter-coated with gold in a Balzers MED 010 unit were analyzed for SEM by a JEOL JSM 6010LA electron microscope (High Equipment Centre, University of Tuscia, Viterbo, Italy) and by an ESEM FEG XL30 (Philips) under low vacuum conditions (at the University of Insubria, Como, Italy). SEM image analyses were performed using the free-license software ImageJ with the Diameter J 1-018 Plugin to measure nanoparticle sizes (100 measurements for each type).

### 2.9. Atomic Force Microscopy (AFM)

The nanostructured layer topography was characterized by atomic force microscopy (AFM; Nanosurf Flex-AFM, version 5—C3000, Liestal, Switzerland), which captured images of the layer surface in tapping mode using 190 Al-G tips, 190 kHz, and 26 N/m over image areas of 400 and 100 µm^2^ (20 × 20 and 10 × 10 µm, respectively). Topography images were processed using Gwyddion© 2.64 software.

### 2.10. Transmission Electron Microscopy (TEM)

Transmission electron microscopy (TEM) micrographs were captured at 200 keV using a transmission electron microscope equipped with an analytical double-tilt holder. After removal from the collector, the electrospun nanofibers were placed onto Lacey–Carbon copper TEM grids by simple adherence. A Zeiss LIBRA200FE microscope (Carl Zeiss AG, Oberkochen, Germany) was used to carry out TEM analyses. ImageJ software with the Diameter J Plugin was used to measure MIP- and NIP-NPs’ pore sizes (over 100 samples).

### 2.11. Focused Ion Beam (FIB)

A fiber sample deposited onto an aluminum substrate was cut using a dual-beam (Gallium ion beam) Tescan GAIA 3 FIB-SEM to produce a sample section and ultra-thin lamella. The lamella was then checked and analyzed by a TEM Thermo Scientific Talos F200X (Thermo Fisher Scientific, Waltham, MA, USA).

### 2.12. Electrical and Sensing Measurements

The chemiresistors (MIP-NFs/IDEs) were enclosed in a measurement chamber (~1 mL volume) and connected to an electrometer (Keithley 6517, Solon, OH, USA) for powering and recording their electrical outputs, with data transmitted to a PC running LabVIEW 2014 software (National Instruments, Austin, TX, USA). Clean air (5.0, Nippon gases) was used to record the current under controlled humidity and temperature conditions, applying potential values ranging from −4.0 to +4.0 V. The resistance (R) of the fibrous coating, within the linear range, was calculated using Ohm’s Law.

Dynamic sensor measurements were conducted at 25 °C by applying a +3 V potential between the interdigitated electrodes. A four-channel MKS 247 system was used to control up to four MKS mass flow controllers (MFCs) with flow rates set between 0 and 200 sccm. Pure air, humidified through a Nafion© tube inside a sealed glass jar saturated with water vapor, modulated the carrier gas’s humidity through the system. Relative humidity and temperature were measured using a HIH 4602 sensor (Honeywell International Inc., Charlotte, NC, USA ). Incoming air was mixed with known concentrations of terpenes (S-Lim, Lin, and α-Pin). Both clean and vapor-saturated air streams converged in a 10 mL mixing chamber before entering the measurement chamber. Analyte flow passed through a gas bubbler containing the analytes before mixing with the carrier flow and entering the measurement chamber. Each measurement was initiated after fully recovering the baseline current under clean airflow. IDE responses were calculated as ΔI/I_0_, where ΔI represents the current variation, and I_0_ is the baseline current obtained during the airflow cleaning process. A scheme of the measurement setup is depicted in Figure 6A. The recovery and reproducibility of the sensor were evaluated over 8 cycles, alternating the exposure to 70 ppm of S(-)-limonene and the air cleaning at 55% relative humidity (RH). To assess the long-term stability of the MIP-based nanofibrous sensor for S(-)-limonene, some measurements of this monoterpene were conducted over one year.

## 3. Results and Discussion

### 3.1. Particle Characterization

The following section introduces the synthesis process of MIP and NIP nanoparticles, following the step-by-step procedure outlined in the flow chart presented in Figure 1E and described in Section 2.2. The morphology of NIP and MIP nanoparticles was investigated using SEM (Figure 1), which highlighted a spheroidal morphology in both NIP-NPs (Figure 1A) and MIP-NPs (Figure 1B), with low variation in mean diameter (Figure 1C,D and Table 1). Notably, NIP-NPs were approximately 35% smaller than the larger MIP-NPs, indicating that the imprinting process may contribute to an increase in particle size (Table 1). This size difference is likely attributable to the presence of the target molecule during the polymerization process. One plausible explanation for the larger size of MIP-NPs compared to NIP-NPs is that the template molecule acts as a nucleation center during polymerization, guiding the organization of the polymer network. MAA, with its carboxyl group (-COOH), engages in weak non-covalent interactions with limonene. Although limonene lacks conventional hydrogen bond donors or acceptors, the weak polarity of its C=C double bond can interact with the carboxyl group in MAA through weak π–electron interactions. These interactions help orient the template within the polymer matrix, facilitating the formation of specific recognition sites.

During polymerization, the presence of the template molecule (S-Lim) locally affects the crosslinking density, leading to less dense or more spatially expanded regions around the recognition sites. This spatial effect, combined with the physical occupancy of the template within the polymer network, may contribute to the larger size of MIP-NPs compared to NIP-NPs. Thus, the template not only dictates the formation of recognition sites but also affects the overall structure and expansion of the polymer network, resulting in larger MIP nanoparticles. While numerous studies have explored the polymerization of MIP and NIP nanoparticles, the specific impact of template molecules on particle size has yet to be extensively investigated.

Both NIP-NPs and MIP-NPs displayed a rough and porous surface in SEM micrographs (Figure 1A,B). The rough and porous aspect of the MIP nanoparticles primarily resulted from the polymerization conditions and the choice of porogen solvent. Acetonitrile, a highly polar solvent, effectively dissolves MAA and EGDMA, ensuring a homogeneous reaction medium. Its volatility (T_bp_: 81.6 °C) promotes phase separation during polymerization, forming pores in the polymer network. The rate of phase separation influences pore size and distribution: faster rates lead to smaller, more uniform pores, while slower rates produce larger, less uniform pores. At the reaction temperature of 60 °C, below acetonitrile’s boiling point, moderate pore sizes were expected. The AIBN concentration also plays a critical role in controlling the crosslinking rate, thus affecting both the size and distribution of pores. Both MIP- and NIP-NPs displayed tortuous and interconnected cavities. The roughness between MIP-NPs and NIP-NPs looked comparable, with NIP-NPs exhibiting slightly larger fracture widths in NIP (approximately 9.22 ± 3.40 nm for NIP-NPs compared to 7.73 ± 3.70 nm for MIP-NPs; Figure 1B), while MIP-NPs showed a little larger pore sizes (approximately 1.8 ± 0.4 nm for MIP-NPs and 1.4 ± 0.4 nm for NIP-NPs; Figure 2B,D).

TEM analyses of both NIP-NPs and MIP-NPs confirmed the homogeneity in size of these nanoparticles (Figure 2A,C). However, these particles sometimes showed a slight fusion of adjacent spheres (shape inhomogeneity) and frequent apparent polymer leakage at contact points with TEM support grids and between contiguous nanoparticles (Figure 2B,D), underpinning the observed particle aggregation previously described (Figure 1A,B). Both nanoparticle types displayed a “soft morphology” in TEM micrographs, characterized by diffuse or poorly defined edges, likely due to the absence of the ordered lattice structure typical of crystalline materials. Such nanoparticles are generally classified as amorphous [81]. Amorphous materials scatter electrons diffusely, leading to lower contrast and less defined boundaries, contributing to the perception of softness [82]. The irregular and flexible structures of amorphous polymers or crosslinked networks further accentuate this soft morphology, often producing rounded particles. The softness of MIP nanoparticles likely arises from the high molar ratio of MAA (functional monomer) to EGDMA (crosslinker), which reduces network rigidity [83]. For the synthesis of molecularly imprinted nanoparticles (MIP-NPs), the molar ratio of S-Lim:MAA:EGDMA (1:4:8) was applied, following a protocol aimed at balancing polymer network stability with the availability of binding sites for S-Lim [84]. The monomer-to-crosslinker ratio plays a critical role in determining the mechanical properties and stability of the polymer matrix. If the molar ratio favors EGDMA, it leads to a more rigid and mechanically stable structure, which improves the sensor’s durability for long-term use. However, this also reduces the availability of binding sites, potentially lowering sensitivity to the target molecule. At the current molar ratio, the longer alkyl chains of EGDMA appeared to affect the softness of the nanoparticles by (i) introducing flexibility due to their ability to rotate and bend in the amorphous regions, (ii) lowering the crosslinking density because of reduced efficiency, and (iii) creating less defined surface boundaries in the nanoparticles [85]. The softness of the nanoparticles, linked to the flexibility of the polymer matrix and its ability to undergo conformational changes, enhances their ability to interact with and release target analytes. This makes them particularly suitable for applications such as sensing, drug delivery, and separations. In VOC sensors (e.g., for detecting S-Lim), this property is especially beneficial, as it improves both sensitivity and reusability.

### 3.2. Fiber Characterization

The fiber morphology housing MIP-NPs (MIP-NFs) was first investigated using scanning electron microscopy (SEM). The SEM micrographs of MIP-NFs displayed in Figure 3 depict a complex but ordered three-dimensional network of nanofibers achieved by electrospinning the PVP-MWCNT-MIP suspension onto silicon wafers. The fibers, ranging in diameter from 45 nm to 285 nm, with an average diameter of 134 ± 42 nm (Figure 3C, inset), were well distributed and intertwined, forming a continuous structure. The network demonstrated high interconnectivity, with several junctions and overlapping points (Figure 3A–C). The pores within the network, measured over an area of 5.75 × 3.9 μm, were irregular in both shape and size, as expected in nonwoven and unaligned deposition, with an average pore area of 0.04 ± 0.047 μm^2^. Some fusiform-shaped beads, characterized by the absence of branching and the continuation of the fiber, displayed minor diameters ranging from 0.65 to 3.65 μm (mean diameter: 1.21 ± 0.73 μm). This phenomenon is commonly attributed to high local surface tension or fluctuations in viscosity and electrical charge during the electrospinning process. The presence of MIP-NPs in the polymer matrix may contribute to these irregularities, either because the average fiber diameter is smaller than the one of the MIP-NPs or due to the inhomogeneous distribution of MIP-NPs along the fibers.

To better understand the internal structure of possible polymer beads along the fibers, we used a FIB-SEM to cut a thin transversal lamella (Figure 4A–C). The SEM analysis in (dark) STEM mode [86] (Figure 4D) highlighted several spherical nanoparticles dispersed within the fusiform bead form, suggesting the presence of the MIP-NPs. These particles were also visible in the fiber sections above the bead (see the arrows in Figure 4A,D). Bright-field STEM analysis of the thin lamella (Figure 4E) also confirmed the presence of multi-walled carbon nanotube (MWCNT) segments within the bead matrix.

More detailed topographical information of the PVP/MWCNT/MIP nanofibers (MIP-NFs) at the nanoscale was provided by AFM micrographs (Figure 5A,B) in amplitude mode [87], revealing pronounced surface protrusions. These distinctive features proved the presence of heterogeneous complexes within the fibers, attributed to the incorporation of both MIP-NPs and MWCNTs. The protrusions on the fiber surfaces exhibited irregular sizes and shapes, confirming variations in the distribution of MIP-NPs throughout the polymer matrix. This uneven distribution may contribute to the rough texture of the fibers, enhancing their surface area (Figure 5A,B).

The shape and distribution of MWCNTs are evident in the TEM micrographs (Figure 5C–E). The dispersion of MWCNTs in DMF was optimized through sonication processes in combination with MIP-NPs and PVP solution to ensure a uniform distribution within the electrospun fibers, which is critical for enhancing the sensor’s performance. The nanotubes resulted dispersed throughout the fibers in various configurations: (i) aligned longitudinally with the fiber axis (Figure 5D), (ii) forming spiral structures (Figure 5C), and (iii) wrapping around the fibers in a spring-like arrangement (Figure 5E). Typically, the dispersion state of carbon nanotubes in the electrospinning solution is reflected in their distribution within the fibers [79]. The alignment of MWCNTs within electrospun polymer nanofiber composites is influenced by the forces involved in the formation of nanofibers in the electrospinning process, like electrostatic forces and strong shear forces during the elongation process, as also observed in the case of nanowire alignment within nanofibers [88]. Besides, it is heavily dependent on the stability of the carbon nanotube dispersion in the solution [89]. Factors such as solution viscosity, flow rate, type of collector, solvent, and the interactions among the polymer, nanoparticles, and nanotubes all play significant roles in this process [90]. Furthermore, inherent structural defects, like polymer dislocations and vacancies, may cause irregularities in the configuration of nanotubes in the nanofibers [91]. In addition to all the aforementioned factors, the different configurations observed are also due to the presence of MIP-NPs in the electrospinning solution, which could affect nanotube dispersion all along the nanofibers, thus altering their arrangement therein. An evident example of the possible interaction between MWCNTs and MIP-NPs is the spiral configuration of the MWCNTs displayed in Figure 5C, where the nanotubes seem to tangle around a globose object that appears as an MIP-NP aggregate. Evidence of interactions between MIP-NPs and MWCNTs is also visible in Figure 4E.

### 3.3. Electrical Properties

The measurement setup is illustrated and described in Figure 6A. To quantify and better understand MIP–MWCNT–PVP interactions, current–voltage (I–V) curves were investigated. The MIP- and NIP-NF-coated IDEs’ current–voltage (I–V) curves, carried out at 55% RH and T: 25 °C, are shown in Figure 6B. The plot revealed nonlinear shapes for both the chemiresistors. 

As shown in Figure 6B, when the voltage was increased in the positive direction, the current for both MIP- and NIP-NFs remained steady until it reached a threshold around +1 V, indicating the onset of a charge carrier injection into the nanocomposite. Beyond this point, a sharp rise in current was observed, suggesting the formation of a Schottky barrier at the electrode interface, leading to semiconductor-like behavior with nonlinear characteristics. A similar trend was observed with a negative sign when a negative voltage was applied. At higher positive and negative voltages, the current increased linearly for both IDEs, albeit with different slopes. The resistance values measured at the working potential (+3 V) indicated that both fibrous systems were highly resistive, with R_NIP_ being comparable to R_MIP_ (Table 2), though slightly higher. The sheet resistivity of the fibrous layer was calculated by measuring the electrical resistivity of the device and considering the dimensions of the electrodes, as follows [92]:(1)$$Rs=R*\;\frac{W}{L*np*2}$$
where Rs is sheet resistance (Ω/□), R is electrical resistance (Ω), W is the width of electrode fingers (10 µm), L is the length of electrode fingers (3500 µm), and np is the number of pairs of fingers (120). The results are summarized in Table 2.

The conductivity mechanism in the studied nanocomposites is likely a result of charge transport within the polymer matrix, facilitated by conductive pathways formed by multi-walled carbon nanotubes (MWCNTs) [93]. The incorporation of MWCNTs into the nanofibers improved the electrical conductivity of the sensor. The high surface area and excellent conductive properties of MWCNTs facilitated the formation of a conductive network within the PVP matrix, even at concentrations below the percolation threshold [94]. The PVP matrix acted as a structural scaffold for both the MIP nanoparticles and the MWCNTs. Although PVP is inherently non-conductive, its polar functional groups (e.g., carbonyl groups) can weakly interact with the MWCNT surface, promoting better dispersion and alignment of the MWCNTs within the matrix. This dispersion prevents MWCNT aggregation, ensuring effective integration of their conductive properties into the nanofibrous sensor (Figure 5D). While the MIP nanoparticles themselves are non-conductive, their interaction with MWCNTs can still influence the sensor’s overall conductivity (Figure 4D,E). The reactive surface of MWCNTs allows for weak van der Waals interactions with the MIP nanoparticles, and the graphene-like structure of MWCNTs can engage in π–π stacking interactions with the MAA units of the MIP nanoparticles. These interactions not only support the molecular recognition of specific analytes by the MIP sites but also facilitate electron transport through the MWCNT network. However, since the MWCNT concentration is below the percolation threshold, charge transport is presumably dominated by tunnelling effects [95,96,97,98,99]. Additional factors, such as the formation of a Schottky barrier at the MWCNT–PVP interfaces and imperfections at the nanofiber–electrode boundary, may also influence the observed conductivity.

The I–V curves for the tested materials showed similar shapes, though higher resistance values were recorded for NIP nanofibers, potentially due to differences in nanofiber network density.

The fibers loaded with 1.45 wt.% of MWCNTs exhibited high resistivity, consistent with previous findings on PVP/MWCNT nanocomposites. This behavior aligns with the established percolation threshold of approximately 4 wt.% for similar systems [100,101]. The electrical conductivity of such composites is strongly affected by the distribution of nanotubes within the polymer matrix. When MWCNTs are well dispersed, tunnelling conduction predominates, leading to higher resistivity [97]. Conversely, the formation of MWCNT agglomerates facilitates direct contact between nanotubes, creating additional conductive pathways [96] that enable electrical percolation and electronic conduction [95]. While increasing the amount of MWCNTs improved the conductivity of the system, it also made the sensor less functional in terms of sensitivity. Higher MWCNT content could lead to excessive conductivity, which might not necessarily translate into better sensor performance, as it could interfere with the sensor’s ability to detect and bind the target VOCs effectively. Additionally, to achieve an optimal balance, we selected a 12% PVP solution as the polymer matrix for electrospinning. This concentration provided sufficient structural integrity to form stable nanofibers while ensuring that the fibers were capable to hold MIP-NPs and fairly homogeneous in shape and size. Excessive fiber sizes could limit the sensor’s interaction with VOCs, reducing its sensitivity. Finally, UV-crosslinking of the PVP matrix was performed to improve the structural integrity of the nanofibers. This process enhanced the polymer’s hydrophobicity and mechanical robustness without significantly compromising the sensitivity to VOCs.

### 3.4. Sensor Features

Appendix A provides an overview of the sensing features, such as limitations of some recent sensors developed for detecting VOCs in agriculture. Conversely, Table 3 displays some key parameters of specific limonene sensors based on MIP technology. MIP sensors for limonene detection were developed using MAA as the functional monomer, EGDMA as the crosslinker, and AIBN as the initiator, with a detection limit of 10 ppm and a selectivity of 55% for distinguishing limonene from similar compounds, like limonene oxide and α-pinene [51,52]. Thin films were deposited onto metallic electrodes and heated under nitrogen at 60 °C. A MIP sensor array was also designed to monitor mango ripeness by detecting limonene and other terpenes, like α-pinene and β-pinene [53]. Additionally, MIP-QCM sensors were used to detect limonene in grass and pest-infested trees [54,55]. The solvent used for polymerization (methanol–acetonitrile mixture) influenced the interaction strength and porosity of the polymer films. A capacitive sensor based on MAA, EGDMA, and AIBN in THF was designed to recognize terpenes from mangoes [56]. A capacitive sensor was designed for spin-coating a MIP solution based on MAA, EGDMA, and AIBN, diluted in THF, onto gold IDEs and polymerizing under UV light [57] to recognize selected terpenes released from mango ripeness. Furthermore, a conductive MIP (cMIP) sensor for R(+)-limonene, with a detection limit of 50 ppm was created by blending MIPs with poly(3-hexylthiophene) (P3HT) and using QCM and interdigitated electrodes (IDEs) [58]. Enantiomers of limonene were also detected in essential oils using a pyrrole-based MIP film on electrochemical sensors, achieving a limit of detection of 1.4 × 10^−12^ mol L^−1^ [59]. The UV molecularly imprinted nanofibers of PVP (MINF sensor) achieved a promising LOD of 226 ppb [77]. However, the MINF sensor presented certain limitations, including the need for precise control of the UV exposure time to ensure effective photopolymerization, responsible for the S-Lim binding sites’ distribution.

In our sensor, the molar ratio of S-Lim:MAA (1:4) used to produce the MIP nanoparticles, which are hosted throughout the fibers and have a spherical shape, is expected to provide a higher density of specific binding sites to the sensor. A nanofibrous film is preferred over a bulk sensor because it reduces response times related to the diffusion process, closely mimicking natural structures that are more efficient at absorbing or detecting environmental chemicals. Additionally, the inclusion of MWCNTs boosted conductivity and provided extra surface area for VOC adsorption, enhancing the sensor’s performance.

As expected, the nanostructured MIP-NF sensor exhibited rapid responses to S(-)-limonene, characterized by a swift increase in current, indicating high reactivity (Figure 7A). Notably, the sensor reached a steady state, defined as t_90_ (the time required to achieve 90% of the response), in approximately 180 s at 30 ppm S(-)-limonene under controlled conditions: constant airflow, 55% relative humidity (RH), and a temperature of 25 °C. The response kinetics reflected swift and efficient analyte detection, apparently without hysteresis. Indeed, the sensors consistently returned to baseline levels when exposed to clean air, highlighting their repeatability and reliability (Figure 7A). The calibration curve for the MIP sensor, ranging between 10 and 100 ppm (Figure 7B), indicated that increasing current changes occurred in response to increasing S(-)-limonene partial pressure, according to a nonlinear model, like a Langmuir-type adsorption, with a gradual increase in response that approached saturation only at higher analyte concentrations.

Therefore, sensor sensitivity, defined as the change in response per unit change in analyte concentration, was derived from the slope of the linear region of the Langmuir-like calibration curve (S = ΔI_norm_/C, where ΔI_norm_ is the normalized change in current from its baseline, and C is the analyte concentration). For the MIP-NF sensor, the calculated sensitivity between 10 and 20 ppm (i.e., where the most rapid increase occurred) was S_10-20_: 0.102 ± 0.022 ppm^−1^. LOD (limit of detection) and LOQ (limit of quantification) are two critical parameters for assessing the sensor’s sensitivity and its ability to operate effectively in low-concentration environments. LOD represents the lowest concentration of an analyte that the sensor can reliably detect (Equation (2)), and it is the threshold at which the sensor response can be distinguished from background noise. LOQ, on the other hand, is the lowest concentration at which the sensor can not only detect but also accurately quantify the analyte with acceptable precision and accuracy (Equation (3)). These two parameters were calculated according to:(2)$$LOD=\frac{\left(3*\;\sigma x\right)}{SENS}$$(3)$$LOQ=\frac{\left(10*\;\sigma x\right)}{SENS}$$
where SENS is the sensitivity, α is the slope of the response curve, σx is the standard deviation of the sensor’s baseline noise, LOD is the limit of detection, and LOQ is the limit of quantification. Table 4 presents a comparison of these parameters between the MIP-NF and MINF sensors, with the latter (S-Lim sensor) utilizing the same polymer fibers and conductivity principles (PVP-MWCNTs) but featuring molecular imprinting of the template directly on the fibers.

Comparing the features of the sensor developed in the present study with those of the best-performing sensor previously fabricated and referred to as the MINF sensor, a nearly three times higher sensitivity and a lower limit of detection (LOD) could be observed [77]. This finding highlights that incorporating MIP-NPs specific for S(-)-limonene recognition in the MIP-NF sensor enhanced the detection performance toward this VOC significantly, albeit with a trade-off of increased response time likely due to their structural design.

The MIP nanoparticles in the MIP-NF sensor were engineered to provide a high density of binding sites, accounting for the superior sensitivity. The different sensitivity of the two sensors was reasonably affected by the global number of binding sites for S(-)-limonene molecules present in the two configurations. The higher sensitivity of MIP-NFs vs. MINFs highlighted that a higher number of MIP sites for S(-)-limonene were present in MIP-NFs than in MINFs. The combination of this outcome with that of the lower LOD also suggested a more uniform distribution of these binding sites all along the MIP-NFs. These findings highlighted that incorporating MIP-NPs specific for S(-)-limonene recognition in nanofibers to develop advanced targeted sensors enhanced the detection of this VOC significantly. Notwithstanding the improved performances on MIP-NFs, the path required for S(-)-limonene molecules to reach the binding sites was greater in MIP-NFs and generated a delay in response times (ranging from 30% to 70%, depending on VOC concentration).

About the kind of electrical signal generated by S(-)-limonene adsorption, some authors reported that including limonene as a dopant in a polyelectrolyte matrix enhances ionic conductivity due to interactions between the methyl and methylene groups of limonene (acting as electron donors) and the matrix, which improves ionic mobility and thus increases conductivity [102]. Therefore, adding limonene as an additive not only increases ionic conductivity but also reduces bulk resistance, creating additional conductive pathways within the polymer matrix [103]. In this study, the presence of hydroxyl groups from MAA and EGDMA, along with UV-crosslinking in the air (which may ionize the carbonyl groups of PVP), likely contributed to the observed increase in conductivity in the presence of limonene.

To determine the selectivity and the effectiveness of the sensor to S-Lim, MIP- and NIP-NF-based sensors were exposed to three different terpenes with similar structures. Measurements were performed using the experimental setup shown in Figure 7A. Alpha-pinene and linalool were selected for the similarities in their structures (they are both monoterpenes, i.e., with 10 carbon atoms, and derived from similar building blocks, specifically isoprene units), which make them structurally related, although they differ in specific functional groups (e.g., alcohol group in linalool vs. hydrocarbon structure in limonene and alpha-pinene).

The normalized response of the sensor at 15 ppm of S(-)-limonene was observed to be approximately 4 times higher than the sensor response at the same concentration of linalool and more than 11 times higher than that of alpha-pinene (Figure 8A). The higher response to linalool compared to α-pinene is presumably attributed to the hydroxyl group present in the linalool structure. Both the imprinted polyacrylate and PVP contain polar functional groups that can interact with linalool’s hydroxyl (-OH) group, thereby enhancing nonspecific interactions. These polar interactions may lead to unintended binding of linalool on the sensor surface, even though the sensor is primarily selective for limonene. Consequently, the polar groups in both linalool and the sensor materials increase the likelihood of cross-reactivity, producing a measurable response despite the sensor’s high selectivity for limonene.

An estimation of sensor selectivity [104] among the tested VOCs was described by the selectivity index (SI; Equation (4)):SI(%) = (R_target_/∑R_VOCs_) × 100(4)
where R_target_ is the sensor response to a defined concentration of the analyte and R_VOCs_ is the sensor response to the other chemicals within the measured pattern. The selectivity index of the MIP-NF sensor was 73% for S(-)-limonene when mixed with other terpenes, as depicted in the pie chart of Figure 8B, and it was a little higher than the one of the MINF sensor for the same monoterpene (72%), as reported by Macagnano et al. (2024) [101]. Such a slight increase in selectivity suggested that the nano-in-nano strategy adopted here enhanced the sensor’s analyte recognition properties. Simultaneously, the IDE coated with NIP-NFs provided a very low response to S(-)-limonene (Figure 8C) and to all the other tested chemicals without any apparent selectivity, confirming the efficiency of the imprinting method. Since MIP is designed to lock the limonene molecule in place, it should enhance the sensor’s selectivity and sensitivity by minimizing cross-reactivity with other molecules. The high selectivity of MIP nanoparticles toward S(-)-limonene in the gaseous phase can be attributed to specific interactions between the functional groups of the polymer matrix and the molecular structure of S(-)-limonene. For instance, MAA, with its carboxyl group (-COOH), can engage in hydrogen bonding interactions. While limonene lacks conventional hydrogen bond acceptors or donors, the weak polarity of its C=C double bond enables weak hydrogen bonding with the -COOH groups in MAA. Specifically, the hydrogen atom on the hydroxyl group of MAA can form a hydrogen bond with an electron-rich site on the limonene molecule, such as the π-electrons of the C=C double bond. These interactions, although weak, can contribute to the selective recognition of limonene, especially in a gaseous environment. Although π–π interactions are typically associated with aromatic rings, the conjugated π-system of the limonene C=C double bond can weakly interact with electron-rich sites in the polymer matrix. The carboxyl group of MAA may create localized electron density around the polymer backbone, facilitating these interactions. Furthermore, the spatial arrangement of the polymer network in MIPs allows for potential “π-stacking” interactions between limonene’s double bond and functionalized regions of the polymer. The imprinted cavities in the MIP are specifically tailored to the size and shape of S-Lim molecules, with the described interactions influenced by the steric arrangement of MAA-functionalized sites in the polymer network.

The sensor’s selectivity index was measured at 55% RH. At other humidity levels (see Section 3.5), the sensor exhibited significantly reduced affinity for limonene and slower response times, making it impractical to assess selectivity under those conditions due to the substantial decline in performance.

The recovery and reproducibility of the sensor were evaluated over 8 cycles to 70 ppm. After each cycle, the MIP-NF sensor demonstrated a mean response value of 3.25 with a maximum variation of 5%. The assessment of the long-term stability of the MIP-based nanofibrous sensor for S(-)-limonene showed that the response curve consistently followed that of the MIP-NF sensor measuring S(-)-limonene over short periods, indicating that the MIP-NF sensor maintained stability for at least one year.

### 3.5. Humidity Interference

While this study primarily focused on the response of MIP- and NIP-based nanofibrous sensors under controlled conditions (55% RH, 25 °C), it is essential to consider the impact of varying environmental factors, particularly humidity, on sensor reproducibility and reliability in real-world applications. Environmental humidity is one of the most common interferents in polymer sensors, significantly affecting sensing characteristics, such as response dynamics, selectivity, and sensitivity. Hence, evaluating the sensor’s performance under varying relative humidity conditions is crucial to determining its optimal operating environment.

When comparable volumes of ambient air saturated with water vapor (Figure 9, blue bar) and S(-)-limonene vapors (Figure 9, brown bar) were introduced into the sensor chamber, the results confirmed the sensor’s high selectivity for S(-)-limonene. Only minor increases in conductivity were observed in response to water vapor, with the change being less than 5% of the electrical signal generated by S(-)-limonene vapor.

In the present study, the MIP-NF sensor was tested at 100 ppm of S(-)-limonene under relative humidity levels of 30%, 55%, and 80% to identify the most suitable measuring conditions. At 30% humidity, the sensor exhibited significant noise and a very weak electrical signal. The optimal response was achieved at 55% relative humidity, where the sensor demonstrated superior performance. At 80% relative humidity, although conductivity was three orders of magnitude higher, resulting in less noisy measurements, the ΔI/I_0_ ratio was lower, and response times were slower. Figure 10 illustrates the comparison of response dynamics under these conditions.

The conduction mechanism in the MIP-NF sensor can be attributed to the interaction of polar molecules, such as water, with the PVP polymer. Due to nitrogen atoms in PVP, water molecules are readily absorbed, particularly under higher humidity levels [105]. The hydrophilic nature of polyvinylpyrrolidone (PVP), a key component of the fibrous matrix, significantly contributed to moisture absorption, forming hydration layers within the polymer network. These layers enhanced ionic mobility, which, in turn, affected the sensor’s conductivity. At higher relative humidity (RH > 70%), preliminary observations revealed a notable increase in the baseline current due to the additional ionic conduction facilitated by absorbed water molecules. Conversely, at lower humidity levels (RH < 30%), the sensor’s response showed a marked reduction in sensitivity, likely due to the limited availability of hydration layers necessary for efficient ionic transport. These findings emphasized the importance of calibrating and controlling the sensor for specific humidity ranges to ensure consistent performance. Shinde et al. [106] further supported this mechanism through molecular simulations, demonstrating that significant interactions occurred when PVP chains were arranged so that hydrogen-bonded protons could be shared between adjacent carbonyl groups. If these carbonyl groups were tightly packed, they restricted the mobility of other species and facilitate proton transfer via the Grotthuss mechanism, enhancing the overall conductivity of the sensor. Additionally, the adsorption of S-Lim molecules onto the MIP sites was significantly influenced by ambient moisture levels. Water molecules may compete with the target analyte for binding sites, potentially reducing the sensor’s selectivity. However, the incorporation of multi-walled carbon nanotubes (MWCNTs) helped mitigate this effect by maintaining electronic conductivity even under high-humidity conditions. Figure 10 illustrates the dynamic response of the MIP-NF sensor to varying concentrations of S(-)-limonene at increasing humidity levels (55% and 80% RH), showing that while the sensor remained responsive across these conditions, its sensing performance was optimized at 55% RH. The conductivity mechanisms were influenced by the interaction between the target analyte and the sensor material, as well as environmental factors like humidity. Both ionic and electronic conductivity contributed to the sensor’s performance under different conditions. Under dry conditions, when the target analyte bound to the MIP sites, the contribution to electronic conductivity was negligible because the concentration of MWCNTs was below the percolation threshold, resulting in a barely detectable change in electrical conductivity. As relative humidity increased, more water molecules were absorbed, generating additional H+ ions and shifting the conductivity mechanism toward ion conduction. This shift could be further enhanced by interactions between the target analyte and functional groups in the MIP, which may induce ion exchange or form ionic complexes. Ideally, a balance between ionic and electronic conductivity would enable the sensor to maintain high sensitivity and performance across varying environmental conditions.

In Figure 10, the sensor responses demonstrate environmental humidity’s significant influence on response time and sensitivity. The proposed mechanism is illustrated in the same figure (Figure 10, right). An increase in water ions enhanced conductivity, which impacted the sensor’s sensitivity. However, excessive water vapor saturated the molecular recognition sites, hindering efficient access to S(-)-limonene and leading to nonspecific interactions (refer to the schematic in Figure 10, right). At approximately 55% relative humidity, S(-)-limonene could effectively access these recognition sites. When the sensor was exposed to 100 ppm of S(-)-limonene and the relative humidity decreased from 80% to 55%, there was a marked improvement in performance. The ΔI/I_0_ ratio increased from 1.0 to 3.6, and the response time (t_90_) significantly decreased from 325 s to 135 s.

## 4. Conclusions

This study highlighted groundbreaking advancements in the development and optimization of molecularly imprinted nanofiber sensors for detecting gaseous terpenes, specifically targeting S(-)-limonene as a critical biomarker of plant health. By employing an innovative “nano-in-nano” fabrication strategy that combines electrospinning, molecular imprinting, and conductive nanomaterials, we achieved a sensor with unparalleled performance in terms of selectivity, sensitivity, scalability, and cost-efficiency compared to conventional sensors.

The proposed sensor demonstrated exceptional performance metrics, including a rapid response time (steady state achieved within 200 s) and high sensitivity at ~55% RH. Notable features included a LOD of 190 ppb, a LOQ of 630 ppb, a selectivity index of 73% among similar terpenes, and outstanding reproducibility (±5%). These metrics represent a significant leap forward in detecting biogenic volatile organic compounds (BVOCs), particularly under conditions where traditional VOC sensors struggle with specificity and sensitivity at low concentrations.

The tailored molecular recognition capabilities of molecularly imprinted polymers (MIPs) provided a marked improvement in specificity and selectivity. By integrating MIPs into PVP nanofibers, the sensor’s nanofibrous architecture maximized the surface area available for interaction with target molecules, significantly enhancing analyte interception and overall sensitivity.

The incorporation of multi-walled carbon nanotubes (MWCNTs) further boosted the sensor’s conductivity and stability, while the PVP nanofibers ensured structural integrity, flexibility, and optimal fabric thickness to minimize VOC diffusion limitations. UV-crosslinking enhanced the mechanical robustness of the fibers without compromising sensitivity, resulting in a durable sensor suitable for long-term applications.

To optimize VOC diffusion and environmental stability, molecularly imprinted nanoparticles were meticulously designed with precise molar ratios to achieve an optimal balance of size, structure, and porosity. This design ensured easy accessibility for VOCs while maintaining resistance to environmental stresses, further enhancing the sensor’s reliability.

The advancements introduced in this study make the sensor uniquely suited for monitoring BVOCs in complex environments, where distinguishing specific volatile compounds among numerous others is challenging. In agriculture, this capability is transformative, as real-time detection of stress biomarkers enables early intervention for pest control, nutrient management, and disease prevention. These sensors have the potential to revolutionize crop management by promoting more efficient resource utilization and fostering sustainable agricultural practices.

Moreover, the versatility of the proposed strategy extends beyond agriculture. Tailored molecular recognition sites and tunable polymeric frameworks suggest broad applicability in urban air quality monitoring, industrial contamination detection, and environmental health assessments. Compared to conventional BVOC sensors, the proposed approach provides a robust foundation for advancing precision agriculture, air quality control, and early-warning systems.

While these findings represent a significant step forward, several challenges remain. Environmental stability under fluctuating temperature and humidity conditions requires further attention. Temperature variations can alter the MIP matrix conformation, affecting adsorption/desorption kinetics, while humidity significantly impacts ionic and sensing signals. Potential solutions include incorporating humidity-stabilizing materials (e.g., Nafion^®^ cartridges), implementing compensation algorithms, or encapsulating the sensors with hydrophobic coatings. Protective systems to shield sensors from dust and airborne particulates could also enhance longevity and reliability.

Scaling up production for industrial applications presents another challenge, as ensuring uniform distribution of MIP nanoparticles and MWCNTs within the fibers is critical for reproducibility. Advanced dispersion techniques combined with electrospinning could address this issue, paving the way for cost-effective, large-scale sensor manufacturing.

In summary, the molecularly imprinted nanofiber sensors developed in this study represent a paradigm shift in VOC detection technology, offering unprecedented selectivity, sensitivity, and adaptability. These advancements provide a robust platform for diverse applications, from precision agriculture to environmental monitoring, heralding a new era in VOC sensing.

## Figures and Tables

**Figure 1 polymers-17-00326-f001:**
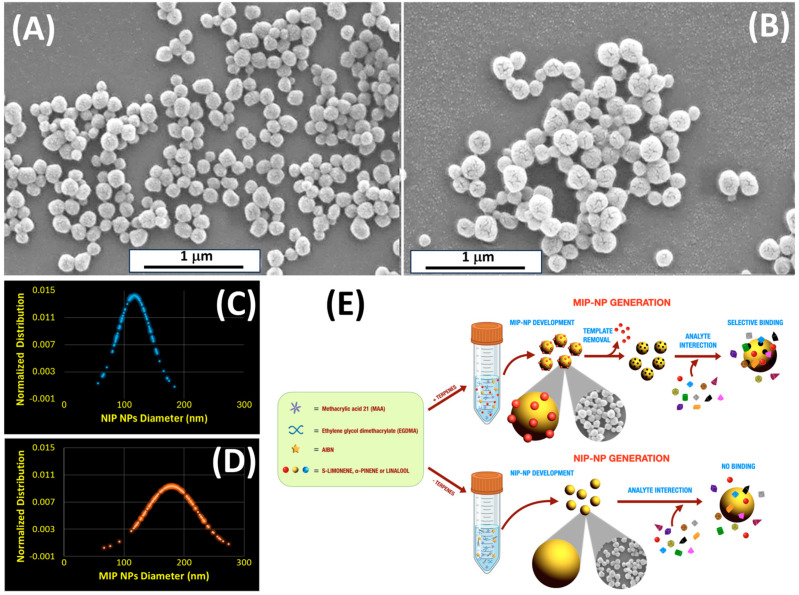
SEM images of the synthesized polymeric nanoparticles. (**A**) Non-imprinted polymer (NIP) nanoparticles, serving as control particles without specific recognition sites, and (**B**) molecularly imprinted polymer (MIP) nanoparticles (NPs), hosting structured recognition sites templated for S(-)-limonene. (**C**,**D**) Plots illustrate the respective normalized distributions of NIP- and MIP-NP diameters calculated from 100 samples. (**E**) Schematic representation of the development of MIP nanoparticles based on MAA and EGDMA, designed for specific terpenes, here, S-Lim. The process also includes the preparation of non-imprinted particles (NIP) as control samples, lacking the molecular imprinting of the target analyte.

**Figure 2 polymers-17-00326-f002:**
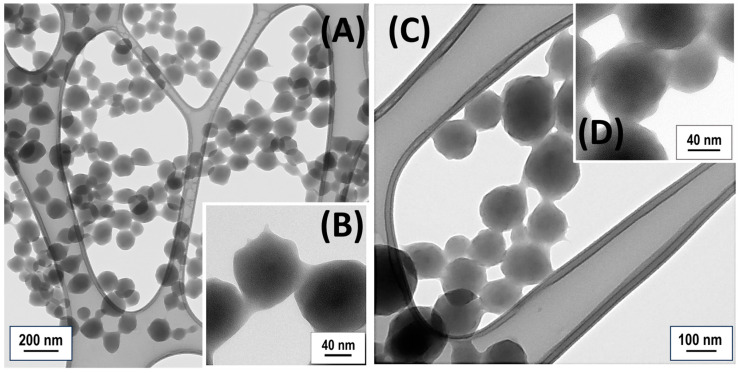
Transmission electron microscopy (TEM) micrographs showing NIP-NPs drop-cast onto a TEM grid at varying magnifications (**A**,**B**) and MIP-NPs (**C**,**D**).

**Figure 3 polymers-17-00326-f003:**
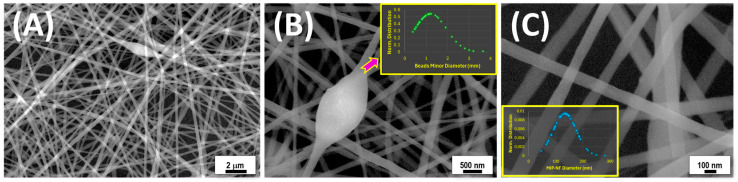
(**A**) SEM images of MIP-NPs collected on a silicon wafer, displayed at progressively higher magnifications in (**B**,**C**). Panel (**B**) highlights a single nanobead structure, while panel (**C**) showcases the details of the nanofiber architecture. The inset in (**B**) presents the normalized minor diameter distribution of approximately 50 beads, whereas the inset in (**C**) illustrates the diameter distribution of 100 nanofibers.

**Figure 4 polymers-17-00326-f004:**
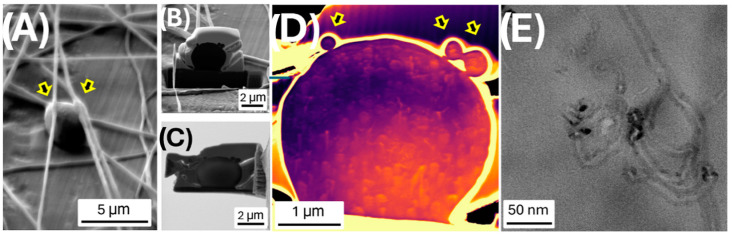
SEM micrographs of (**A**) the region of interest for the FIB cut and milling, (**B**) the first FIB cut, and (**C**) the final lamella. (**D**) False color (to highlight the details) of SEM-STEM image in dark mode and (**E**) STEM (by TEM measurement) of the lamella. Yellow arrows indicate the fibers above the bead and their location in the section.

**Figure 5 polymers-17-00326-f005:**
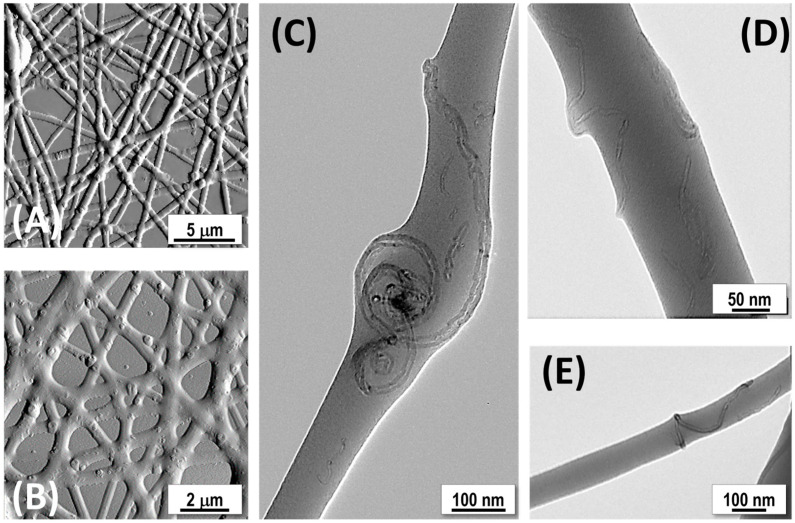
AFM micrographs of PVP/MWCNT/MIP nanofibers (**A**) with a higher-magnification view (**B**). TEM micrographs of the nanofibers with different MWCNT distributions along nanofibers, including spiral configurations (**C**), more aligned structures (**D**), and spring-like arrangements around the polymer (**E**).

**Figure 6 polymers-17-00326-f006:**
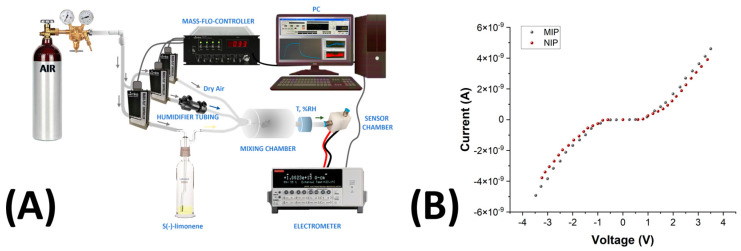
(**A**) The experimental setup used in this study includes an air cylinder and a mass flow controller, with three channels for precise delivery of dry air, humidified air, and template VOCs at specified ratios into a mixing chamber equipped with temperature and humidity sensors, from which the mixture is directed into a mixing and then measuring chamber housing the S(-)-limonene chemosensors. (**B**) Current–voltage (I–V) curves for MIP-NF-based (MIP) and NIP-NF-based (NIP) sensors.

**Figure 7 polymers-17-00326-f007:**
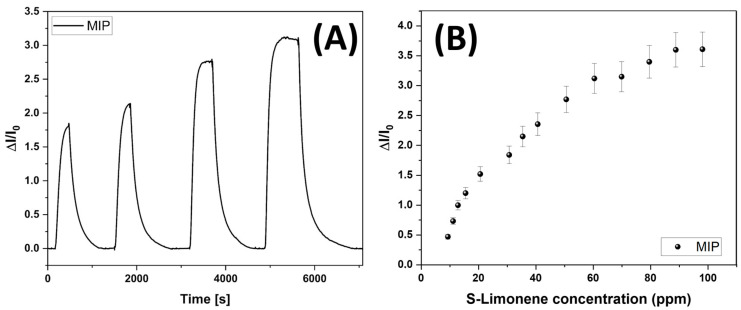
Transient responses of the MIP-NF sensor to increasing concentrations of S-Lim vapors (ranging between 30 and 60 ppm) (**A**). Sensor response curve to S(-)-limonene from 10 to 100 ppm. All data were provided at 55% of relative humidity (**B**).

**Figure 8 polymers-17-00326-f008:**
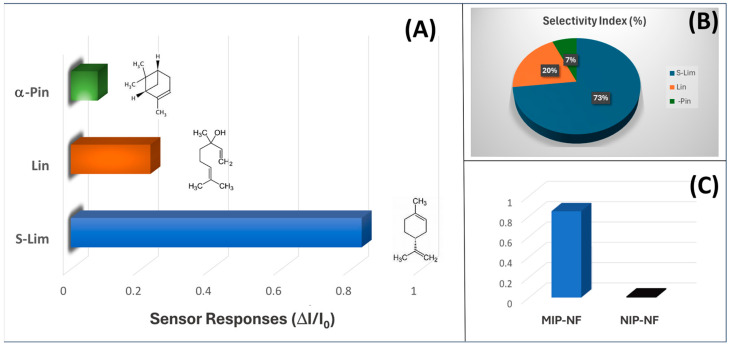
(**A**) Bar plot of the MIP-NF sensor response during exposure to 15 ppm of S(-)-limonene (ΔI/I_0_: 0.800 ± 0.070), linalool (ΔI/I_0_: 0.230 ± 0.020), and alpha-pinene (ΔI/I_0_: 0.077 ± 0.008), and (**B**) pie chart depicting the related sensor selectivity index. (**C**) Bar plot depicting the comparison between MIP-NF- (ΔI/I_0_: 0.800 ± 0.070) and NIP-NF-based sensors during the exposure to 15 ppm of S(-)-limonene.

**Figure 9 polymers-17-00326-f009:**
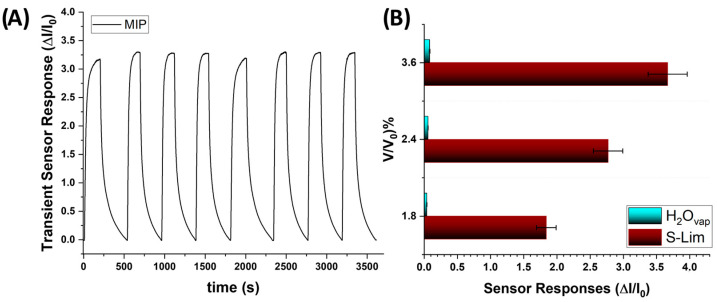
(**A**) Eight cycles of MIP-NF sensor transient measurements to 70 ppm of S(-)-limonene at 55% RH. (**B**) Bar plot of sensor responses to defined volume-flow percentages (1.8%, 2.4%, and 3.6%) of saturated S-Lim (black bar segment) and H_2_O vapors (light-blue bar segment) at 55% RH.

**Figure 10 polymers-17-00326-f010:**
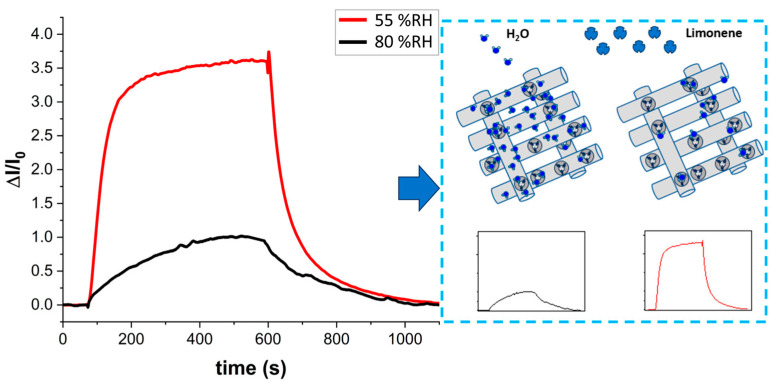
Dynamic response of the sensor when exposed to 100 ppm of S(-)-limonene at 55% and 80% relative humidity, including (on the right) a schematic representation of the potential contribution of H_2_O molecules to S(-)-limonene adsorption onto the nanofiber binding sites.

**Table 1 polymers-17-00326-t001:** Mean diameter of MIP and NIP nanoparticles.

	Mean Diameter (nm)
MIP-NPs	179.40 ± 43.00
NIP-NPs	117.15 ± 28.00

**Table 2 polymers-17-00326-t002:** Electrical properties of MIP-based nanofibers (T: 25 °C; %RH: 55).

	R (Ohm)	W/L (µm)	Rs (Ohm/□)
MIP-NFs	8.35∙10^8^ ± 1.00∙10^8^	2.86∙10^−3^	9.95∙10^3^ ± 1.20∙10^3^
NIP-NFs	1.54∙10^9^ ± 1.85∙10^8^	2.86∙10^−3^	18.35∙10^3^ ± 2.30∙10^3^

**Table 3 polymers-17-00326-t003:** A selection of the most-cited studies on MIP-based sensors specifically targeting limonene detection.

Type	Molar Ratio	Sensing Layer	Transducer	Linear Range (ppm)	LOD (ppm)	Reference
T:Styrene:DVB	0.06:1:1.5	Film	QCM	20–250	20	[54]
T:MAA:EGDMA	1:5:20	Film	QCM	-	-	[53]
T:MAA:EGDMA	1:4:20	Film	QCM	300–2100	7.43	[56]
T:Styrene:DVB	0.06:1:1.5	Film	QCM/IDE		50	[58]
T:MAA:EGDMA	1:4:20	Film	QCM	1–1000	-	[52]
T:MAA:EGDMA	1:4:20	Film	QCM		10	[51]
MAA:EGDMA	1:5:20	Film	IDE	1–400	-	[57]
PAA:PVP:MWCNT	1:4:8	Nanofibers	IDE	1–60	0.23	[77]

**Table 4 polymers-17-00326-t004:** Comparison of figures of merit among MIP-NF and MINF sensors.

Sensors	SENS (ppm^−1^)	LOD (ppb)	LOQ (ppb)	Reference
MIP-NF sensors	0.102 ± 0.022	190	630	This study
MINF sensors	0.037 ± 0.001	226	-	[77]

## Data Availability

The raw data supporting the conclusions of this article will be made available by the authors upon request.

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
