# Peer review of "Cutting-Edge Sensor Design: MIP Nanoparticle-Functionalized Nanofibers for Gas-Phase Detection of Limonene in Predictive Agriculture"

_polymers, 2025, doi:10.3390/polym17030326_

Round 1
Reviewer 1 Report
Comments and Suggestions for Authors
I would like to thank the authors for devoting efforts to write this interesting paper However, here are few suggestions to further improve the paper:
1- The methodology and fabrication is not described in full details. If authors have reported the methods previously in prior publications can reference to that or add the details such as solvent used for PVP solution or conversion MIP to nanoparticles after synthesis.
2- The number of measurements for nanoparticles and nanofibers and the number of SEM images used to be mentioned in the paper plus the softwares used.
3- The produced fibers ranged between 45 to 285 micron which can be lower than nanoparticle diameter to almost double the size. when the particles are embedded deep inside the fibers, do they have any contribution to sensor activity or only those protruding out contribute?
4- The authors should add data on the sensors selectivity performance under different relative humidity conditions since it had a significant impact on the sensitivity performance.
Author Response
Dear Reviewer,
Thank you for your valuable feedback.
Please find the responses to your comments below and in the attached PDF file. We hope our explanation provides clarity and addresses your concerns effectively.
Best regards,
Antonella Macagnano

Reviewer 2 Report
Comments and Suggestions for Authors
The paper shows some significant findings on the particular subject area. However, there are several things need to be improved as such:
- The fundamental problem of this work is it addresses—precision in volatile organic compound (VOC) detection—has competing approaches. Highlight how this method outperforms established techniques in practical agricultural applications.
- Clarity on the optimization process for parameters like monomer-to-crosslinker ratio, MWCNT distribution, and their effects on sensor performance is needed. Did the authors explore trade-offs between sensitivity and structural robustness?
- The paper need to include statistical significance or error bars for calibration curves and transient responses.
- Provide comparative data against industry-standard sensors (not just MIP-based ones).
- Discuss the impact of environmental factors such as varying humidity levels in greater detail beyond Figure 10, ensuring reproducibility in real-world conditions.
- A deeper theoretical discussion on the interaction mechanisms such as hydrogen bonding or π-π interactions between S(-)-limonene and the polymer matrix are required to be elaborated
- The role of MWCNTs is clear in enhancing conductivity, but their interaction with MIP nanoparticles and the matrix needs more quantification.
- Details on real-world deployment such as environmental extremes, reusability after contamination are limited. Address practical challenges, including scaling production and maintaining uniform MIP distributions.
- Broader comparisons to sensors used for similar agricultural applications would provide a more comprehensive assessment.
- While the focus is on agricultural stress biomarkers, potential applications in environmental VOC monitoring or industrial contamination detection are mentioned briefly. Elaborating on these broader implications would increase the study's impact.
- Clarify the contributions of ionic versus electronic conductivity mechanisms in different operational scenarios.
Comments on the Quality of English LanguageImprove the language
Author Response
Authors sincerely thank the Reviewer for his/her effort and thoughtful insights.
Authors have carefully addressed all his/her observations, and their responses are detailed in the revised manuscript (red types). Here, comments from the Reviewer are marked as Q followed by the corresponding number, while our replies are labeled A with the same numbering for clarity.
Sincerely
Antonella Macagnano

Reviewer 3 Report
Comments and Suggestions for Authors
Dear Authors
The Research Article is very Impressive and It represents a significant contribution to the new science world. I have provide some comments and kindly request that you address them as much as possible . Thank you and best of luck .

English is clear. But please eliminates any potential confusion caused by longer sentences.
Author Response
Dear Reviwer,
Thanks for your comments and questions. We have appreciated the chance to contribute to clarify our work.
We uploaded a Word file replying to all your requests: in red, in bold, and in quotation marks, you will find the text that has been also added to the manuscript.
Kind regards
Antonella Macagnano
